# Advancing Dermatological Diagnostics: Interpretable AI for Enhanced Skin Lesion Classification

**DOI:** 10.3390/diagnostics14070753

**Published:** 2024-04-02

**Authors:** Carlo Metta, Andrea Beretta, Riccardo Guidotti, Yuan Yin, Patrick Gallinari, Salvatore Rinzivillo, Fosca Giannotti

**Affiliations:** 1Institute of Information Science and Technologies (ISTI-CNR), 56124 Pisa, Italy; andrea.beretta@isti.cnr.it (A.B.); rinzivillo@isti.cnr.it (S.R.); 2Department of Computer Science, Universitá di Pisa, 56124 Pisa, Italy; riccardo.guidotti@unipi.it; 3Laboratoire d’Informatique de Paris 6, Sorbonne Université, 75005 Paris, Italy; yuan.yin@isir.upmc.fr (Y.Y.); patrick.gallinari@sorbonne-universite.fr (P.G.); 4Faculty of Sciences, Scuola Normale Superiore di Pisa, 56126 Paris, Italy; fosca.giannotti@sns.it

**Keywords:** Explainable Artificial Intelligence, skin image analysis, dermoscopic images, adversial autoecnoders, AI in healthcare

## Abstract

A crucial challenge in critical settings like medical diagnosis is making deep learning models used in decision-making systems interpretable. Efforts in Explainable Artificial Intelligence (XAI) are underway to address this challenge. Yet, many XAI methods are evaluated on broad classifiers and fail to address complex, real-world issues, such as medical diagnosis. In our study, we focus on enhancing user trust and confidence in automated AI decision-making systems, particularly for diagnosing skin lesions, by tailoring an XAI method to explain an AI model’s ability to identify various skin lesion types. We generate explanations using synthetic images of skin lesions as examples and counterexamples, offering a method for practitioners to pinpoint the critical features influencing the classification outcome. A validation survey involving domain experts, novices, and laypersons has demonstrated that explanations increase trust and confidence in the automated decision system. Furthermore, our exploration of the model’s latent space reveals clear separations among the most common skin lesion classes, a distinction that likely arises from the unique characteristics of each class and could assist in correcting frequent misdiagnoses by human professionals.

## 1. Introduction

Decision support systems powered by Artificial Intelligence (AI) have recently seen a significant surge in interest across various fields due to their impressive capabilities. Nonetheless, their application in sensitive areas, particularly those affecting human decisions such as in healthcare, has sparked ethical concerns regarding the opacity of AI-driven decisions [1,2]. There is an emerging consensus on the need for AI systems that not only augment the decision-making process of medical professionals with AI-generated insights and recommendations [3,4] but also ensure that the rationale behind AI decisions is transparent. This is especially pertinent in the context of skin image classification, where the lack of interpretability in the decision-making process of deep learning models complicates the interaction between the AI system and medical practitioners. It is, therefore, crucial to enhance existing classification models with explainability features that facilitate more insightful interactions and provide additional diagnostic tools [5]. This paper addresses these challenges within the scope of skin lesion diagnosis from images.

The flourishing field of Explainable AI (XAI) has thus gained considerable attention [2,6,7,8], with saliency maps being a prevalent form of explanation for image classifiers. These maps visually represent the contribution of each pixel toward the model’s decision, offering a pixel-level insight into the decision-making process. Despite the array of approaches to generate saliency maps, they are often criticized for their lack of clarity in critical medical situations. Explanation methods vary, being classified as model-specific or model-agnostic based on their reliance on the inner workings of the AI model, and as global or local, indicating whether the explanation pertains to the model as a whole or to individual predictions [6,7]. Notable model-specific explainers like IntGrad [9], GradInput [10], and ε-LRP [11] specialize in deep neural networks and generate saliency maps. However, these maps can be fragmented and challenging to interpret in urgent medical scenarios. Conversely, model and data agnostic local explainers such as lime [12] and shap [13] suffer from their reliance on image segmentation, which can compromise the plausibility of their explanations by reducing them to obscured versions of the original image, a practice that diminishes trust and utility in medical contexts [14].

Addressing these limitations, abele (Adversarial Black box Explainer generating Latent Exemplars), was proposed as a local, model-agnostic explainer tailored for image classifiers [15]. abele explains decisions by providing exemplar and counter-exemplar images—those classified similarly or differently from the input image, respectively—alongside a saliency map that underscores decision-critical areas.

This paper aims to build upon and refine the methodologies discussed in [3,15,16,17], exploring the application of an explanation method in a genuine medical context, specifically for diagnosing skin lesions from images. Utilizing the labeled dataset from the ISIC 2019 (International Skin Imaging Collaboration) challenge (https://challenge.isic-archive.com/data/, accessed on 1 September 2019), we train a cutting-edge deep learning classifier based on the ResNet architecture [18] and elucidate the model’s decisions using abele [15]. This approach allows practitioners to interpret the model’s reasoning through the provided exemplars and counter-exemplars. Our study evaluates the utility of these explanations through a user study with medical professionals, novices, and laypeople, aiming to gauge their impact on trust and confidence in the AI system.

The contributions of this paper are multifaceted. Firstly, it refines and evaluates abele within a real-world medical scenario. Secondly, it introduces a latent space analysis performed by the adversarial autoencoder in abele, providing insights that could aid in distinguishing between similar skin lesion types commonly confused by human diagnosticians. Thirdly, it develops a user interface for exploring abele’s explanations. Lastly, through a comprehensive user study, it demonstrates that explanations enhance trust and confidence in AI decision systems, particularly among domain experts and highly educated individuals. This study also uncovers a persistent skepticism toward AI among older demographics, as well as a decrease in trust among experts following incorrect AI advice. Additionally, we observe that the saliency maps generated by abele are superior to those by other local explainers, such as lime and lore [19].

The remainder of this document is structured as follows: Section 3 outlines the methodology, Section 4 details the case study and introduces the visualization tool, Section 5 introduces the visual explanation outputted by the explainer, Section 6 presents the survey findings, Section 8 explores the latent space analysis, and Section 10 summarizes our findings and suggests avenues for future research.

## 2. Related Work

XAI has emerged as a focal point in medical imaging, aiming to shed light on the intricate workings and decisions of AI models in a clear and comprehensible manner. In the healthcare sector, the role of XAI is pivotal for enhancing the trust and confidence of both practitioners and patients toward AI-driven diagnostic and treatment approaches.

Numerous research efforts have delved into the application of XAI within medical imaging contexts, encompassing areas such as chest X-rays [20], CT scans [21], and MRI scans [22]. These studies have employed a range of XAI techniques, including but not limited to saliency maps, attribution maps, and decision trees. A landmark study by Jampani et al. [23] marked one of the initial attempts to utilize saliency map models across various medical imaging domains. Subsequent research has expanded upon this foundation, exploring decision trees and rule-based systems for articulating explanations behind AI-driven diagnoses in medical imaging. Notably, Seung et al. [24] demonstrated the effectiveness of decision trees in elucidating deep learning model predictions for chest X-ray analyses. This particular study underscored the capability of decision trees in offering transparent explanations regarding AI model decision-making processes.

Moreover, XAI has been rigorously evaluated in diagnostic scenarios involving breast cancer [25], lung nodules [26], and brain tumors [27], among others. Across these evaluations, a diversity of XAI methodologies—including saliency maps, decision trees, and attribution maps—have been leveraged to illuminate the decision-making mechanisms of AI models.

Within the field of medical imaging, deep learning has emerged as a pivotal force, particularly in the detection and segmentation of skin lesions [28]. Recent advancements in convolutional neural networks (CNNs) have significantly enhanced the accuracy and efficiency of diagnosing various skin conditions, including melanoma, one of the deadliest forms of skin cancer [29]. Studies such as that by Esteva et al. (2017) [30] have demonstrated the capability of deep learning models to match or even surpass the diagnostic accuracy of dermatologists. Moreover, the integration of deep learning techniques with dermatoscopic imaging has opened new avenues for automated analysis, enabling the detailed examination of skin lesion features with unprecedented precision [31]. These technologies not only aid in early detection but also play a crucial role in delineating the boundaries of lesions, facilitating accurate surgery and treatment planning.

Collectively, these investigations affirm the critical and innovative role of XAI in medical imaging, showcasing its potential to foster greater trust and confidence in AI-assisted diagnoses and to enhance the understanding of the AI models’ operational dynamics.

Despite these advancements, XAI in medical imaging confronts several pressing challenges that warrant attention [32], such as issues related to trust, data bias, interpretability, privacy, and integration into clinical workflows, among others. Key challenges include:Building Trust: A significant hurdle lies in engendering trust toward AI systems and their outputs. Achieving a high level of system explainability is vital for fostering trust and facilitating clinical adoption.Addressing Data Bias and Interpretability: The presence of bias in medical imaging data and the complexities associated with data interpretation can exacerbate when AI algorithms are trained on such data, potentially leading to skewed outcomes.Ensuring Privacy: The sensitive nature of medical imaging data necessitates stringent protections. Concerns regarding the handling and storage of these data, alongside the risk of data breaches or misuse, are paramount.Overcoming Data Limitations: The scarcity of high-quality medical imaging data can impede the efficacy of XAI algorithms, posing challenges to model training and validation.Clinical Workflow Integration: Seamlessly incorporating XAI systems into existing clinical workflows demands a thorough assessment of algorithmic performance and limitations, as well as their potential impact on clinical decision-making processes.Compliance with Regulation and Standards: The XAI domain in medical imaging is governed by a complex web of regulations and standards. Ensuring compliance with these regulatory frameworks is both challenging and labor-intensive.

This work aims to develop a human-centric approach to tackle existing hurdles in Explainable Artificial Intelligence for medical imaging. Our goal is to facilitate the integration of these technologies into clinical settings by developing dependable and transparent decision-support tools that incorporate XAI techniques into the clinical workflow.

Our approach to XAI is categorized under generative explanation-based methods. This involves leveraging a generative mechanism to craft visual explanation. Specifically, the Contrastive Explanations Method (CEM) [33] is adept at generating explanations by identifying the minimal necessary or absent regions in an image for a particular classification decision. Concurrent research [34,35,36] has focused on generating explanations that highlight modifications needed to either amplify or diminish the classifier’s confidence in its prediction, employing concepts such as prototypes or counterfactuals. The technique of Explanation by Progressive Exaggeration [37] introduces a novel way to elucidate the decisions of opaque classifiers by utilizing a generative adversarial network (GAN) [38]. It systematically alters the input in a manner that shifts the model’s prediction, functioning in a model-agnostic fashion and relying solely on the predictor’s values and its gradient relative to the input.

Our work aligns with these cutting-edge explorations; however, we opt for a distinctive architecture, the adversarial autoencoder (AAE). The AAE presents a key advantage over the GAN by offering a more nuanced manipulation of the latent space—the compact, lower-dimensional representation of data. This precision facilitates the production of samples more aligned with the actual data distribution, ensuring coherence and relevance in the generated outputs. Furthermore, unlike GANs, which are predicated on a min-max contest between the generator and discriminator, AAEs incorporate a reconstruction loss, enhancing the similarity between generated and input data.

Additionally, AAEs have utility in unsupervised representation learning, enabling them to distill a condensed representation of data useful for other tasks like classification. This capability stems from the encoder component of the AAE, which projects data into the latent space, and the decoder, which reconstructs data from this space back to its original form.

In summary, AAEs offer a more controlled and interpretable mechanism for data generation compared to GANs. This makes them a superior and more versatile instrument for XAI applications, particularly within the context of medical imaging.

## 3. Methods

In this section, we present the two main components of the methodology adopted to classify and explain the dataset. More details can be found in [3,15,18].

### 3.1. Black Box Classifier

To establish a robust classifier that excels in image classification tasks and supports subsequent learning phases, we opted for the ResNet architecture. Renowned for its proven efficacy across a number of complex datasets and challenges [18], ResNet stands out as our architecture of choice. Rather than training a ResNet model from scratch, we opt for a transfer learning set, utilizing a ResNet model pre-trained on the ImageNet dataset. This approach is particularly beneficial in scenarios where data availability is limited relative to the complexity of the network [39]. During the transfer learning process, we replaced the model’s final fully connected layer with a new one, tailored to match the dataset’s class count. Consequently, this classification layer undergoes training from the beginning, while the remaining parts of the ResNet model are fine-tuned. We adopted binary cross entropy loss for each class, framing the task as a series of one-vs-rest binary classification challenges.

### 3.2. Adversarial Autoencoders

A key concern when utilizing synthetic examples for black box explanation development is ensuring consistency with the original examples’ distribution. Addressed in [15], this challenge is met through the deployment of adversarial autoencoders (AAEs) [40], a hybrid model that combines the principles of generative adversarial networks (GANs) [38] with autoencoder-based representation learning.

AAEs, as probabilistic autoencoders, are designed to generate new items that closely resemble the input training data. They achieve this by aligning the aggregated posterior distribution of input data’s latent representation with a chosen prior distribution. The AAE model encompasses an encoder mapping from $R^{n}$ to $R^{k}$, a decoder mapping back, and a discriminator assessing the authenticity of the latent features, where *n* is the image pixel count and *k* represents the latent feature count. Within this framework, *x* represents an instance of training data, with *z* denoting its latent representation derived via the encoder. The AAE is characterized by various distributions, including the prior p(z), the data $p_{d}\left(x\right)$, the model p(x), and the encoding and decoding functions q(z|x) and p(x|z). The goal is for the aggregated posterior q(z), obtained through encoding, to mirror the prior distribution p(z), ensuring fidelity in the generated examples while minimizing reconstruction errors. Through this process, the AAE model successfully confuses the discriminator, making it challenging to distinguish between genuine and generated latent instances (see Figure 1).

The AAE training process encompasses two stages: Reconstruction, focusing on minimizing the loss between encoded and decoded data, and Regularization, which fine-tunes the discriminator using both real training data and encoded values. Upon completion, the decoder acts as a generative model, bridging the prior distribution p(z) with the data distribution $p_{d}\left(x\right)$.

### 3.3. ABELE Explainer

abele (Adversarial Black box Explainer generating Latent Exemplars) is a local, model-agnostic explainer tailored for image classifiers [15]. For a given image *x*, abele gives explanations comprising sets of exemplars and counter-exemplars, as well as a saliency map. These exemplars and counter-exemplars are images classified identically or differently to *x*, respectively, offering insights into the classification rationale. The saliency map further delineates regions influencing the image’s classification. abele initiates the explanation process by generating a latent feature space neighborhood via an adversarial autoencoder (AAE) [40], followed by learning a decision tree to provide local decision-making and counterfactual rules [41]. This process involves selecting and decoding exemplars and counter-exemplars conforming to these rules to produce a saliency map.

#### 3.3.1. Encoding

The image *x* in question is encoded through the AAE, with the encoder yielding a latent representation $z\in R^{k}$, utilizing *k* latent features, where *k* is significantly less than *n*, the dimensionality of *x*.

#### 3.3.2. Neighborhood Generation

abele generates a set *H* comprising *N* latent feature space instances, resembling *z*’s characteristics. This neighborhood, inclusive of instances mirroring b(x)’s decision ($H_{=}$) and those diverging ($H_{\neq }$), aims to replicate *b*’s local behavior. The generation of *H* may follow diverse strategies, with our experiments favoring a genetic algorithm to optimize a fitness function [41]. Post-generation, each h∈H undergoes validation and decoding through the disde module, subsequently being classified by the black box model *b* to ascertain its class *y*.

#### 3.3.3. Local Classifier Rule Extraction

With the local neighborhood *H* established, abele constructs a decision tree classifier *c*, training it on *H* labeled according to $b(\tilde{H})$. This surrogate model seeks to closely emulate *b*’s behavior within the defined neighborhood, extracting rules and counterfactual rules. This method, illustrated in Figure 2, encapsulates the journey from the initial image to the derivation of the decision tree, highlighting the extraction of pivotal rules. This process is denoted as llore, a latent variant of lore [41].

#### 3.3.4. Explanation Extraction

In contexts such as medical or managerial decision-making, explanations often revolve around referring to exemplars with similar (or differing) decision outcomes. abele adopts this rationale, modeling the explanation of an image *x* as a triple $e=\langle {\tilde{H}}_{e},{\tilde{H}}_{c},s\rangle$, comprising exemplars ${\tilde{H}}_{e}$, counter-exemplars ${\tilde{H}}_{c}$, and a saliency map *s*. Exemplars and counter-exemplars represent images with outcomes that match or differ from b(x), respectively. abele generates these through the eg module (Figure 3-left), initially producing a set of latent instances *H* that fulfill the decision rule *r* or counter-factual rules Φ, as depicted in Figure 2. Subsequently, it validates and decodes these into exemplars ${\tilde{H}}_{e}$ or counter-exemplars ${\tilde{H}}_{c}$ via the disde module (see Figure 4). The saliency map *s* underscores regions in *x* influencing its classification or steering it toward a different category. This map is derived using the se module (Figure 3-right), which calculates the pixel-to-pixel difference between *x* and each exemplar in $\tilde{H}e$, assigning the median of these differences to each pixel in *s*. Formally, for each pixel *i* in the saliency map *s*, it is defined as $s\left[i\right]={\mathrm{median}}_{\forall {\tilde{h}}_{e}\in {\tilde{H}}_{e}}\left(x\left[i\right]-{\tilde{h}}_{e}\left[i\right]\right).$

The efficacy of abele heavily relies on the quality of the encoder and decoder functions used; the more effective the autoencoder, the more realistic and valuable the explanations become. The following section delves into the autoencoder’s structural nuances necessary to achieve reliable outcomes for the ISIC dataset.

### 3.4. Progressively Growing Adversarial Autoencoder

We detail here the customization of abele we conducted in order to make it usable for complex image classification tasks.

#### 3.4.1. Challenges of Generative Models

Training generative adversarial models presents significant challenges, often marred by various common failures such as convergence issues and the well-documented Mode Collapse [42]—a situation where the generator produces a limited variety of outputs. These issues stem from the adversarial training dynamic, where the generator and discriminator are trained simultaneously in a zero-sum game, making equilibrium elusive. As the generator improves, the discriminator’s performance may deteriorate, leading to less meaningful feedback and potentially causing the generator’s performance to falter if training continues beyond this point.

Moreover, real-world datasets, particularly in healthcare, compound these difficulties with issues like fragmentation, imbalance, and data scarcity, which hamper the efficiency and accuracy of machine learning models, especially fragile generative models.

A standard training approach for an AAE on the ISIC dataset, without addressing these issues, led to subpar performance, primarily due to mode collapse. To mitigate these generative failures and dataset limitations, we implemented a suite of cutting-edge techniques capable of addressing these challenges and enabling successful AAE training with satisfactory performance.

#### 3.4.2. Addressing Mode Collapse in Generative Models

A diverse output range is desirable for generative models; however, the generator’s tendency to favor outputs that appear most plausible to the discriminator can lead to repetitive generation of a single or a limited set of outputs, known as mode collapse. Various techniques, such as Mini Batch Discrimination [43], Wasserstein Loss [44], Unrolled GANs [45], and Conditional AAE [46], have been proposed to alleviate mode collapse by either implementing empirical fixes or adjusting the training scheme’s internal structure.

The exact causes of mode collapse and other failures are not fully understood, with such phenomena becoming more frequent in the health domain, likely due to challenges like limited data, the necessity for high-resolution image processing, and unbalanced datasets. To successfully train an AAE and avoid these pitfalls, we adopted a combination of techniques tailored to address these specific challenges.

#### 3.4.3. Progressively Growing Adversarial Autoencoder

Progressively Growing GANs, as introduced in [47], extend the GAN training process to foster more stable generative model training for high-resolution images. This technique begins training with low-resolution images, gradually increasing resolution by adding layers to both the generator and discriminator models until reaching the target size.

In typical GAN setups, the discriminator evaluates the generator’s output directly. However, in an AAE framework, the discriminator assesses the encoded latent space rather than the reconstructed image. To harness the benefits of progressive growth in this context, we designed the Progressively Growing Adversarial Autoencoder (PGAAE). This approach starts with a basic convolutional layer block in both the encoder and decoder to reconstruct low-resolution images (7 × 7 pixels). Incrementally, we add more blocks, enhancing the network’s capacity to process the desired image size (224 × 224 pixels), while keeping the latent space dimensions constant. This ensures that the discriminator consistently receives inputs of the same size. While fixing the discriminator’s network may seem advantageous, we discovered that gradually expanding its width allows for processing increasingly complex information more effectively. Conversely, deepening the discriminator tends to destabilize training, leading to various failures, including performance degradation and catastrophic forgetting [42].

The rationale behind this methodology stems from the training instability caused by complex, high-dimensional data. Generating high-fidelity images challenges generative models to replicate both structural complexity and fine details, where high resolution exacerbates discrepancies, undermining training stability. Additionally, large images necessitate significant memory, reducing the feasible batch size and introducing further training instability.

Layer-by-layer learning enables the model to initially grasp broad structural aspects before refining focus on detailed textures. This progressive layer introduction acts as a sophisticated form of regularization across both encoder and decoder networks, smoothing the parameter space to mitigate issues like mode collapse.

The PGAAE network paradigm is reported in Figure 5. The structure begins with a simple AAE, focusing on 7 × 7 pixel images, progressively advancing through six stages to achieve 224 × 224 pixel image reproduction. To enhance the discriminator’s capability at each stage, its architecture broadens, incorporating two dense layers that progressively expand from 500 to 3000 neurons. Each convolutional block consists of a conv2d layer followed by batch normalization and a ReLu activation, with either max pooling or up sampling depending on the encoder or decoder role.

#### 3.4.4. Denoising Autoencoder

A significant challenge in training generative models is their tendency to learn the identity function, particularly when hidden layers surpass input nodes, allowing simple data replication without meaningful representation learning.

Denoising autoencoders, as described in [48], introduce stochasticity to combat this by corrupting input images, which the model then strives to reconstruct. This process not only deters the model from identity mapping but also fosters the learning of robust representations.

Introducing noise to the discriminator’s inputs [49] further enhances generalization, mitigates the vanishing gradient issue, and fosters better convergence. Combining denoising techniques with noise injection into the discriminator enhances the model, particularly noticeable in achieving high-quality reconstructions in latent spaces with 256 features. Gaussian noise with a standard deviation of σ=0.1 proved effective, with little benefit observed beyond this range.

#### 3.4.5. Mini Batch Discrimination

Mini Batch Discrimination, as detailed in [43], was introduced to prevent the generator network from collapsing. This technique discriminates across entire minibatches of samples within generative adversarial networks, rather than evaluating individual samples.

The principle behind this approach is for the discriminator to assess batches of data in their entirety, rather than single data points. This strategy significantly simplifies the identification of mode collapse, as the discriminator can recognize when samples within a batch are excessively similar and thus should be deemed inauthentic. Consequently, this compels the generator to diversify its output within each batch. An $L_{1}$ penalty norm is integrated with the input and directed toward the penultimate layer of the discriminator, quantifying the similarity among samples within the same batch. This penalization prompts the discriminator to reject batches that exhibit high internal similarity. Coupled with the progressive growth of the network, this technique has proven effective in averting mode collapse in batches of moderate size (16–64), albeit at the expense of a slight increase in discriminator complexity. Training with smaller batch sizes, necessitated by hardware constraints due to the processing of high-resolution images and managing a high-dimensional latent space, inherently raises the risk of mode collapse.

As per [43], the fine-tuning of two hyperparameters, designated as *B* and *C*, is required for the minibatch discrimination layer. These parameters determine the number of discrimination kernels and the dimensionality of the space for calculating sample closeness, respectively. Theoretically, larger values of *B* and *C* enhance performance but at the cost of computational efficiency. An optimal balance between accuracy and computational speed was found with B=16 and C=5.

#### 3.4.6. Performance

Following an extensive optimization of the encoder, decoder, and discriminator architectures, our PGAAE equipped with 256 latent features achieved a reconstruction error, quantified by the RMSE, ranging from 0.08 to 0.24. This variance is contingent upon whether the data pertain to the most prevalent or the rarest class of skin lesions. Employing data augmentation was crucial for addressing the dataset’s limitations regarding scarcity and imbalance. This strategy substantially mitigated mode collapse, enabling the generation of diverse and high-quality skin lesion images. Consequently, the ABELE explainer is now proficient in producing coherent and meaningful explanations.

It is important to underscore that the ABELE framework operates effectively irrespective of the classifier employed. Naturally, different classifiers yield distinct explanations, highlighting the flexibility and adaptability of the ABELE system.

## 4. Case Study

This section outlines our case study, including the characteristics of the training dataset and the methodologies employed for training both the black box classifier and the autoencoder.

### 4.1. Dataset

The International Skin Imaging Collaboration (ISIC), under the auspices of the International Society for Digital Imaging of the Skin (ISDIS), launched the Skin Lesion Analysis toward Melanoma Detection Challenge aiming to bolster global efforts in melanoma diagnosis (https://challenge2019.isic-archive.com/, accessed on 1 September 2019). This challenge focuses on developing a classifier capable of distinguishing among nine distinct diagnostic categories of skin cancer: MEL (melanoma), NV (melanocytic nevus), BCC (basal cell carcinoma), AK (actinic keratosis), BKL (benign keratosis), DF (dermatofibroma), VASC (vascular lesion), SCC (squamous cell carcinoma), and UNK (unknown, none of the others/out-of-distribution), see Figure 6. The provided dataset comprises a training set with 25,331 images of skin lesions labeled by category and a test set containing 8238 images, the labels of which are not publicly accessible.

### 4.2. Black Box Training

From the training set, we allocated 80% of the samples for training and the remaining 20% for validation purposes. The UNK category was excluded from our training focus since it is not present in the training set, aligning with our objective to develop a diagnostic classifier. The necessity for a reliable classifier to assist medical practitioners influenced this decision, endorsing a model capable of rejecting UNK samples when confidence in classification is low. This capability is crucial from a diagnostic accuracy perspective, preferring a cautious rejection of out-of-distribution samples over potentially erroneous labeling. Due to varying image resolutions, we employed specific preprocessing techniques:For training, images undergo random scaling, rotation, and cropping to fit the network input, ensuring the lesions remain undistorted. The processed images are resized to 224 × 224.Validation and test images are first scaled to 256 × 256 based on the shorter edge, then centrally cropped to 224 × 224.

Evaluation followed the original challenge’s metric system, utilizing normalized multi-class accuracy as the performance measure. This metric, defined as the average recall across all classes, ensures equal importance across categories, preventing biased performance toward dominant ones. The optimized ResNet model, selected based on validation set performance, achieved a balanced multi-class accuracy of 0.838 on the test set (see Table 1 for the full performance description). Given the controlled conditions under which images were captured, ensuring minimal distributional shifts between training and validation sets, cross-validation was deemed unnecessary. To circumvent overfitting given the dataset’s size, we fine-tuned a pre-selected pre-trained ResNet model, basing our architectural choice on its historical efficacy and computational feasibility rather than dataset-specific performance. The learning rate was fine-tuned to $10^{-4}$, optimizing both convergence speed and validation accuracy, with the best-performing model on the validation set being retained for further use.

The 50-layer ResNet architecture comprises 18 sequential modules, including one conv1 module (7 × 7, 64 filters, stride 2), three conv2 modules, four conv3 modules, six conv4 modules, three conv5 modules, and a final fully connected module fc. Each conv2 to conv5 module constitutes a residual block with three convolutional layers, where the block’s output is a sum of its input and the convolutional output. Spatial reduction occurs only in the first layer of conv3, conv4, and conv5 modules, with the fc module serving as the classification layer. The fc module integrates average pooling followed by a 9-output fully connected layer with sigmoid activation to make predictions across the different diagnostic categories. The innovative structure of ResNet, particularly through its residual blocks, facilitates the training of deeper networks by addressing vanishing gradients. This architecture ensures that the network can learn complex patterns associated with various skin lesion types while maintaining computational efficiency.

Each residual block within the conv2 to conv5 modules consists of a specific sequence of convolutional layers designed to process and enhance the feature representation of the input images. These blocks employ a combination of 1 × 1 and 3 × 3 convolutional filters, allowing the network to capture both the detailed and abstract features of skin lesions effectively. By strategically increasing the number of filters and adjusting the stride in these blocks, the network progressively refines its feature maps, leading to a more discriminative representation suitable for classification tasks.

The decision to incorporate a newly trained prediction layer (fc module) at the end of the network underscores our commitment to tailoring the model to the specific requirements of skin lesion classification. This approach allows for fine-grained tuning and adaptation to the unique characteristics of the ISIC dataset, ensuring that the model is well-equipped to handle the variability and complexity inherent in dermatological imaging.

In summary, the deployment of a 50-layer ResNet architecture, coupled with thoughtful preprocessing and strategic model tuning, forms the cornerstone of our approach to tackling the challenge of skin lesion analysis toward melanoma detection. By leveraging the robustness and depth of ResNet, alongside a dataset-specific training regimen, we aim to push the boundaries of automated medical diagnosis, offering a tool that augments the capabilities of healthcare professionals in their fight against skin cancer.

### 4.3. PGAAE Training

Adapting abele for the sophisticated image classification task tackled by the ResNet black box classifier necessitated specific customizations. These adjustments are detailed in [3]. After extensive optimization across all three network components (encoder, decoder, and discriminator), our Progressively Growing Adversarial Autoencoder (PGAAE) with 256 latent features attained a root mean square error (RMSE) spanning from 0.08 to 0.24. This range reflects variability in error based on the frequency of occurrence of the skin lesion classes under consideration. The choice of 256 latent features was determined through initial testing, revealing it as the optimal balance between achieving satisfactory reconstruction accuracy, maintaining high image resolution, and conserving computational resources. Within the context of processing images of the targeted 224 × 224 resolution, a latent feature count ranging between 64 and 512 is typically advocated in the literature.

Data augmentation played a crucial role in addressing the challenges posed by dataset scarcity and imbalance. This strategy significantly diminished the occurrence of mode collapse, enabling the generation of diverse and high-quality skin lesion images (Figure 7). Equipped with PGAAE, abele demonstrates its capability to furnish insightful explanations, as verified through a participant survey discussed in the subsequent section.

### 4.4. User Visualization Module

This segment introduces the innovative visualization module for interpreting explanations rendered by abele. The module elucidates the black box model’s recommendations alongside the explanations generated by abele. A screenshot from a dedicated web application (https://kdd.isti.cnr.it/isic_viz/, accessed on 1 March 2023) illustrates the module’s functionality, displaying the analyzed image, the black box’s classification, and a synthetically generated counter-exemplar by abele in the upper section of the interface. For instance, an image of melanoma and its counter-exemplar, classified as melanocytic nevus, are showcased in Figure 8.

The lower part of the application presents a collection of images akin to the analyzed instance, bearing the same classification. This neighborhood, curated by abele and depicted as a list, offers insights into the latent space’s variance surrounding the examined image. The counter-exemplar—distinguished from the original instance by its different classification—is chosen from this collection based on its minimal Euclidean distance to the original image in the latent space, yet with a maximized prediction for an alternate label. Additionally, the module’s bottom section exhibits four exemplars; these are images abele generated, all sharing the black box’s assigned label to the original image.

The abele visualization module, developed in JavaScript as a web application, interfaces with a backend through a RESTful API, enabling interaction with the black box and abele. A demonstrator version was created, allowing users to select from a predefined set of instances for exploration, rather than uploading new ones. This demonstrator served as a foundation for conducting the survey highlighted in the following discussion.

## 5. Explanation

The interface designed to convey the outcomes of both the classifier and the explanator to users features a streamlined visual layout organized into four distinct sections: (1) the original image analyzed by the CNN, alongside the classification it was assigned; (2) a highlight section that underscores specific areas of the image that positively (depicted in brown) or negatively (depicted in green) influenced the classification decision; (3) a collection of synthetic prototypes created by the AAE that share the same classification as the input image; (4) a counter-exemplar, which is a synthetic image representing a prototype assigned a different classification by the CNN compared to the input image.

An illustrative example presented in Figure 9 depicts an image classified as Melanocytic nevus. The highlighted section allows users to discern which portions of the image were deemed significant by the CNN for its classification. This result is further elucidated by showcasing four prototypes: images synthesized by the AAE, designed to bolster the user’s confidence in the black box’s decision by facilitating a comparison between the original image and the exemplars. The counter-exemplar serves to challenge the black box’s conclusion by offering an image akin to the input yet classified under a different category by the CNN.

ABELE compiles statistics related to the input’s neighboring instances within the latent space. These data aid in understanding how the CNN’s model space segments around the specific input, providing insights into the range of classifications the black box associates with the given instance. For the instance illustrated in Figure 9, the latent space statistics and rules are encapsulated as follows:$$\begin{array}{cc} \; & \mathrm{Neighborhood}\{NV\;:\;41,\;BCC\;:\;18,\;AK\;:\;4,\;BKL\;:\;26,\;DF\;:\;11\}\\ \; & e=\left\{\begin{array}{c} \mathrm{rules}=\{7>-1.01,\;99\leq 0.07,\;225>-0.75,\;255\leq -0.02,\\ \;\;\;238>0.15,\;137\leq -0.14\}\to \{\mathrm{class}:\;\mathrm{NV}\},\\ \mathrm{counter}-\mathrm{rules}=\{\{7\leq -1.01\}\to \{\mathrm{class}:\;\mathrm{BCC}\}\} \end{array}\right. \end{array}$$

Here, the Neighborhood section outlines the distribution of synthetic latent instances produced by the AAE. Rules and counter-rules are delineated in relation to the ordinal positions within the latent space. While this format is primarily for internal purposes and does not directly inform the user, it serves as a foundation for the visual interface to facilitate an interactive enhancement of the provided explanation.

Another explanation, showcased in Figure 10, refers to a basal cell carcinoma case. A counter-exemplar depicting a vascular lesion (VASC) is crafted in alignment with a local counterfactual rule derived from the decision tree.

The comprehensive breakdown of latent rules for this case is as follows:$$\begin{array}{cc} \; & \mathrm{Neighborhood}\{NV\;:\;34,\;BCC\;:\;24,\;DF\;:\;28,\;VASC\;:\;14\}\\ \; & e=\left\{\begin{array}{c} r=\{7\leq 0.81,\;187\leq 0.46,\;224\leq -0.10,\;219\leq 0.07,\\ \;\;242>-0.25\}\to \{\mathrm{class}:\;\mathrm{BCC}\},\\ c=\{219>0.07\}\to \{\mathrm{class}:\;\mathrm{VASC}\} \end{array}\right. \end{array}$$

## 6. Validation and Assessment

This study presents a method designed to evaluate the effectiveness of abele explanations in the context of skin lesion diagnosis. Our primary objective was to ascertain the value of these explanations in aiding physicians and healthcare professionals in the diagnosis and treatment of skin cancers. Additionally, we sought to gauge their confidence in diagnosis models based on opaque algorithms and the clarity of the explanations offered by the explainer tool.

### 6.1. Survey Design and Methodological Approach

The survey was structured around ten questions, each associated with a distinct medical case image. The format for each question was uniform across all cases, consisting of four segments. The study protocol commenced with participants enrolling online and giving their informed consent. They were then provided with a concise overview of the task at hand. The method of manipulation was direct, aligning with methodologies adopted in prior studies [50,51]. Through this research, our goal is to deepen the understanding of the dynamics influencing the preference for automated guidance in medical contexts. Furthermore, we aim to build upon the existing literature by examining the potential repercussions of suboptimal automated advice and its impact on system trust. Unless otherwise noted, each segment presented a new image *x* to the participants. Each question was identified by a sequential number, denoted as Q*i*, where *i* ranges from 1 to 10. Each question, from Q1 to Q10, incorporates the same four points, from P1 to P4, as detailed subsequently.

Point 1 (P1). Participants were shown an unlabeled skin lesion image randomly selected from the dataset, alongside its abele-generated explanation, as displayed by the visualization module. Specifically, two exemplars and two counter-exemplars from another lesion class were shown to the participants. They were then asked to categorize the image into one of two predefined classes using the explanation as a guide. This segment is aimed at determining whether the explanations provided by abele significantly aid in differentiating between images, even for those without expert knowledge. From another perspective, this serves as a practical assessment of the usefulness metric, which has been theoretically evaluated in [15].

Point 2 (P2). Participants were shown a labeled image and were requested to assess their confidence in the classification made by the opaque algorithm (using a 0–100 slider).

Point 3 (P3). The same labeled image from P2 was presented again, but this time accompanied by the explanation generated by abele. Participants were asked to re-evaluate their confidence after reviewing the explanation. The purpose of P2 and P3 is to ascertain whether exposure to an explanation leads to a change in confidence in the AI’s capabilities.

Point 4 (P4). Participants were asked to evaluate the extent to which the exemplars and counter-exemplars assisted them in aligning their classification with the AI’s decision, and their trust in the explanations produced by abele.

Throughout the survey, participants were not made aware of the accuracy of their predictions nor were they allowed to revise their prior responses or explanations upon receiving new information. To examine how participants react to incorrect advice, question six (Q6) intentionally included a misclassification (P2), followed by misleading advice regarding exemplars and counter-exemplars (P3). The remaining nine cases featured images that were correctly classified. Each survey question, encompassing the points described above, was designed to evaluate different facets of the participant’s interaction with the AI-generated explanations, assessing the four segments across all ten questions.

### 6.2. Hypotheses and Goals

The research framework was structured around specific hypotheses aimed at evaluating the impact of abele explanations on skin lesion classification tasks. These hypotheses are as follows:H1: The explanations provided by abele facilitate the classification task for the users, particularly for domain experts, who are expected to achieve higher classification accuracy (assessed implicitly through P1).H2: The explanations generated by abele enhance users’ trust and confidence in the classifications made by the black box model (assessed implicitly through P2 and P3, and explicitly through P4).H3: Participants exhibit a significant decrease in confidence and trust in the model after being presented with incorrect advice (assessed implicitly through the deliberate introduction of an error).

### 6.3. Results and Discussion

The survey was completed by 156 participants. These individuals enrolled in the survey via an online platform, after which they digitally acknowledged a consent form, completed a demographic questionnaire, and received an overview of various types of skin cancers involved in the study. To ensure meaningful data analysis, only responses from participants who completed at least one entire question (10% of the questionnaire), covering all four points, were considered. The collective demographic profile of the participants is illustrated in Figure 11. Notably, 94% of the participants came from a scientific background, with 27% having educational achievements in medicine or dermatology.

Initially, we evaluated the participants’ ability to classify skin lesion images using the explanations by assigning scores based on their performance in P1. Participants were categorized into two groups: Sub-sample A, consisting of those who correctly classified at least 70% of the images, and Sub-sample B, comprising the remainder. Alternative thresholds within the [60–90%] range were considered but ultimately not used, as they did not significantly alter the statistical outcomes. The overall average performance was impressive (82.02%), including among those without specialized knowledge in the domain (78.67%), and was even more pronounced among those with a medical or dermatology specialization (91.26%). An analysis using the one-way ANOVA on ranks (Kruskal–Wallis H test) [52] between sub-samples A and B revealed that educational level or age did not significantly affect classification performance. However, there was a notable difference (F=4.061, p=0.043) based on the participants’ fields of specialization—those with a medical or dermatology background were more prevalent in sub-sample A, supporting H1 for domain experts and highlighting commendable performance among participants from other fields.

Figure 12 displays the change in participants’ confidence in the black box classification before and after exposure to abele explanations, reflecting responses from P2 and P3. An increase in trust after viewing exemplars and counter-exemplars was observed in all questions except Q3 and Q9, indicating that explanations generally bolster model trust. Notably, the anomaly in Q3 suggests that non-medical experts influenced this particular outcome. A significant boost in confidence from 67.69% to 77.12%, peaking for Q6 (+21.95%), was noted—Q6 being uniquely misclassified by the black box model and showing the lowest pre-explanation confidence at 53.08%. This trend may indicate a resistance to incorrect advice, where consistent erroneous suggestions actually restore confidence levels. Participants are initially resistant to incorrect advice, but consistency in such advice resets their perception. This observation aligns with prior studies in the field of algorithm aversion [50,51,53].

Confidence increases were not uniform across demographics. Figure 13 highlights several insights: an observable rise in confidence across all age groups except those over 55, whose confidence is inherently low and diminishes further post-explanation, possibly due to a generational distrust in AI technology. Educational level inversely correlates with initial confidence levels, yet post-explanation confidence surges, particularly among the more educated—a hint at the Dunning–Kruger effect [54]. Predictably, individuals with medical backgrounds expressed greater confidence than their counterparts from other scientific or non-scientific fields.

Specifically designed to explore H3, Q6 focused on responses to misclassifications by the black-box model. Data indicate a modest skepticism toward the black box’s sixth classification, with no significant drop in confidence after incorrect advice (68.75% for Q1 to Q5, 60.03% for Q6, and 66.71% for Q7 to Q10). Yet, a more focused analysis on medical experts reveals a 14% confidence decline following incorrect advice (78.04% for Q1 to Q5, 56.19% for Q6, and 63.95% for Q7 to Q10), corroborating H3: domain experts’ trust and confidence in the model diminish after encountering inaccurate advice.

Figure 14 condenses the impact of exemplars and counter-exemplars on the recognition of lesion classes as perceived by participants in P4. Consistent with confidence trends observed around Q6, both experts and non-experts reported a drop in confidence in the assistance provided by exemplars and counter-exemplars. Notably, abele’s explanations were found to be more beneficial for medical experts than the general population, with exemplars proving more influential than counter-exemplars for all groups. This effect could stem from the task’s classification complexity, where the relevance of exemplars escalates with the number of classes, diverging from binary classification tasks where exemplars and counter-exemplars may hold similar weight.

The validation survey, encompassing a diverse demographic of participants, has unearthed intriguing findings that both reinforce and extend the existing literature in the field. Notably, our observations align with the discourse in recent studies like that of Ribeiro et al. [12] on model-agnostic interpretations and that of Lundberg and Lee [13] on SHAP values, which advocate for the customization of explanations to suit the user’s expertise. A particularly compelling observation from our survey was the differential impact of explanations on diagnostic confidence across varying levels of expertise. Prior work, such as that by Holzinger et al. [55], emphasized the importance of XAI in enhancing user trust and understanding; our findings introduce a nuanced perspective where the specificity and nature of explanations may significantly influence user responses. Specifically, while domain experts showed increased confidence when provided with detailed, technical explanations, laypersons and novices were more positively influenced by simplified, visually driven explanations. This dichotomy underscores the necessity of adapting explanation models to the user’s background, echoing the sentiments of Gunning and Aha [56] regarding the adaptability of XAI systems. Such insights advocate for the development of adaptive XAI systems capable of dynamically adjusting the complexity and format of explanations based on user profiles, leveraging user feedback to optimize explanatory output. This advancement could pave the way for personalized explanation frameworks, as discussed by Caruana et al. [57], enhancing the interpretability and accessibility of AI-assisted diagnostics for a wider audience, thus marking a significant stride toward democratizing medical diagnostic tools through AI.

## 7. Explaining via Saliency Map

In Figure 9, we present an example of abele’s explanation mechanism. Synthetic exemplars and counter-exemplars provided by abele prove to be significantly more informative than traditional saliency maps. These maps allow for a comparison with those generated by established explanatory frameworks such as lime and lore. The saliency maps depicted in Figure 15 yield deletion AUC (Area Under Curve) scores of 0.888 for lime, 0.785 for lore, and 0.593 for abele. The deletion metric evaluates the decline in the probability of the designated class as crucial pixels, as identified by the saliency map, are incrementally removed from the image. A lower AUC score suggests a more effective explanation. This metric was calculated across a dataset of 200 images, with the average scores indicating that segmentation-based methods (lime: 0.736 mean AUC score, lore: 0.711 mean AUC score) tend to underperform in generating meaningful saliency maps, whereas abele excels by producing more detailed maps (0.461 mean AUC score). In Figure 16 (Top), we outline the deletion curves, represented as the mean AUC of accuracy versus the percentage of pixels removed for 200 sample images. Notably, abele’s curve descends more swiftly and begins at an earlier point relative to the percentage of pixels removed, indicating a more refined and detailed saliency map.

A parallel observation is made when analyzing the insertion metric, which adopts an opposite strategy by incrementally adding each pixel according to its ascending importance. This process expects an improvement in the black box prediction as more features are incorporated, leading to a stepwise increase in model performance. A larger AUC indicates a superior explanation. Figure 16 (Bottom) illustrates that abele achieves consistently higher insertion scores across an average of 200 samples (0.417 for lime, 0.471 for lore, and 0.748 for abele). The rapid ascent in abele’s insertion curve signifies that its saliency map more accurately identifies the image segments most critical to the classifier’s decision-making process.

## 8. Explaining through Latent Space Analysis

The PGAAE model employed by abele maps the ISIC dataset into a 256-dimensional latent space, exhibiting a structured posterior distribution. This feature, as highlighted in [15], enables the use of latent space for visualizing the proximity among individual data instances, offering valuable insights. Such visualizations can assist medical experts and data scientists in better comprehending the distinctive characteristics of various skin cancers, thereby enhancing classification accuracy or trust in the explainer.

Employing multidimensional scaling (MDS) [58] for dimensionality reduction, we translate the latent space information into a two-dimensional visual representation. This process converts the 256-dimensional latent space into a 2D visual field. Figure 17 visualizes the latent encoding of eight skin cancer classes, revealing that primary features of skin lesions can also be discerned in this 2D projection. Notably, all classes except melanoma distribute around the perimeter, avoiding the center, whereas melanomas predominantly occupy the central region of the plot.

This distribution pattern suggests a similarity hierarchy among the skin cancer types. As noted in [59], Benign Keratosis is often misdiagnosed as melanoma, with misidentification rates ranging from 7.7% to 31.0% across different studies. To further explore differentiation capabilities among skin lesion classes, a Random Forest (RF) classifier with 500 trees was trained on the 2D MDS space. This classifier successfully distinguishes Melanoma from Benign Keratosis with 85.60% accuracy (see Figure 18-left), providing a visual tool for distinguishing between lesion types with performance comparable to that of the original complex model. Melanocytic Nevus also exhibits unique characteristics, with a significant proportion of samples centralized in the plot, reflecting the clinical observation that a considerable number of melanomas, especially in younger patients, evolve from benign nevi [60]. The RF classifier demonstrates a 78.53% accuracy in distinguishing Melanoma from Melanocytic Nevus (see Figure 18-right), paralleling the accuracy of state-of-the-art classification techniques.

Our study extends beyond the traditional two-dimensional latent space analysis, introducing a novel application of multidimensional scaling (MDS) in three dimensions. This enhancement is not merely a technical increment but a strategic move to unlock deeper insights into the complex nature of skin lesions. By transitioning from a 2D to a 3D latent space representation, we aim to explore the nuanced interplay between various lesion types, potentially uncovering hidden patterns and relationships that were previously obscured.

Our exploration revealed a distinct spatial distribution of two prevalent skin lesion classes—melanoma and melanocytic nevus—when modeled in this enriched three-dimensional space. Specifically, we observed once again a robust separation between these classes, with melanomas predominantly located within a spherical region, marked by red dots, and melanocytic nevi, denoted by green dots, primarily positioned outside this spherical boundary (see Figure 19). This spatial arrangement not only validates the model’s capability to distinguish between these clinically significant categories but also suggests a deeper, perhaps previously unexplored, biomarker-based differentiation between them.

Moreover, the 3D mapping facilitated a more intuitive and comprehensive visualization of the data, offering a tangible, three-dimensional landscape for medical practitioners to navigate. This approach could significantly enhance diagnostic accuracy by providing clinicians with a more detailed, spatially nuanced understanding of the lesions under examination. Importantly, this method allows for the identification of outliers or atypical presentations, which are often crucial in early-stage melanoma detection.

To further enrich this diagnostic landscape, in future works we will propose the integration of exemplars and counter-exemplars within this 3D model. By zooming into specific data points—individual lesions in this context—we can offer detailed comparative analyses between similar lesion types. This not only aids in the immediate diagnostic process but also serves as an educational tool, enabling practitioners to refine their diagnostic criteria based on visual and spatial comparisons of lesion characteristics in the latent space.

Given the intense focus on melanoma detection within oncological research, accurately predicting the transformation of a nevus into malignant melanoma remains a significant challenge. Future studies should consider the temporal evolution of oncological data. Our methodologies and discoveries could aid clinicians in more precisely evaluating the potential malignancy of benign skin lesions.

## 9. Discussion and Future Work

In light of the significant advancements made in the field of dermatological diagnostics through interpretable AI, this section outlines new challenges and proposes practical experiments to further the research direction initiated by this study. Our objective is to bridge existing gaps and pave the way for groundbreaking solutions in the realm of medical diagnostics.

Despite the successes achieved, a challenge remains in generating high-fidelity explanations for complex or rare skin lesion cases where the model’s confidence is low. To address this, we propose an experiment involving the creation of a more diverse and challenging dataset that includes underrepresented skin lesion types. This dataset will be used to train and evaluate an enhanced version of ABELE, focusing on its ability to generate meaningful explanations for these complex cases.

Another promising direction involves the exploration of cross-modal explanation methods that not only leverage visual explanations but also incorporate textual descriptions generated through natural language processing techniques. This experiment aims to develop a multimodal explanation framework that provides clinicians with a holistic understanding of the AI’s decision-making process.

Experiment 1a: Integrate ABELE with a language model capable of generating descriptive explanations for the visual exemplars and counter-exemplars.Experiment 1b: Conduct user studies with medical professionals to assess the impact of multimodal explanations on their trust and understanding of AI-based diagnostics.

To further enhance the practical utility of AI in dermatology, we propose the development of a real-time interactive explanation system. This system may allow users to interactively query the AI model about specific regions of interest in the skin lesion images and receive instant visual and textual explanations.

The rapid evolution of deep learning architectures offers opportunities to improve the accuracy and interpretability of skin lesion classification models. We propose an experiment to explore the application of novel neural network architectures, such as Transformer models, in the context of dermatological image analysis.

Experiment 2a: Evaluate the performance of Transformer-based models on the ISIC dataset and compare it with the ResNet-based approach.Experiment 2b: Investigate the integration of Transformer models with ABELE to assess the impact on explanation quality and user trust.

Finally, we advocate for a longitudinal study to monitor the adoption and impact of AI and XAI tools in dermatological practice over time. This study may focus on understanding the evolving needs of healthcare professionals and how AI tools can be adapted to meet these requirements.

The proposed challenges and experiments aim to push the boundaries of current research in interpretable AI for skin lesion classification. By addressing these challenges, we hope to unlock new possibilities for AI-assisted diagnostics that are more trustworthy, paving the way for broader acceptance and utilization in clinical settings. Exploring novel architectures, multimodal explanations, and real-time interaction systems promises to enhance the diagnostic process, enabling clinicians to make more informed decisions with greater confidence in the AI’s recommendations. Through the proposed longitudinal study, we can gain valuable insights into the long-term effects of AI integration in dermatology, identifying areas for improvement and adaptation to ensure that AI tools remain relevant and beneficial in the face of evolving medical practices and patient needs.

## 10. Conclusions

This paper demonstrates the application of classification and post hoc explanation methodologies in the context of skin lesion detection. It has been established that abele, following meticulous customization and training, can generate insightful explanations that significantly benefit medical practitioners, offering superior qualitative value over traditional local explainers. The primary challenge lies in the generative model’s training phase. Latent space analysis reveals an intriguing distribution of images within the latent space, potentially aiding in the differentiation of commonly misclassified skin lesions (benign versus malignant). Furthermore, a survey involving experts and non-experts in skin cancer and healthcare sectors corroborated the hypothesis that unvalidated explanation methods lack utility. Future research avenues include applying abele to various diseases and health domains, particularly those reliant solely on raw images or scans of specific body parts. In skin cancer diagnosis, tactile feedback plays a crucial role alongside visual analysis. Enhancements to the user visualization module to support real-time explanation generation would necessitate substantial efforts and resources, as explanation extraction currently depends on the complexity of the image.

Despite the advancements presented in our study, it faces certain limitations that provide avenues for future research. One primary limitation is the reliance on a predefined dataset, which may not fully represent the diversity of skin lesions and different shades of skin color encountered in clinical practice. This can potentially lead to biases in model training and explanation generation. To address this, future work could focus on expanding the dataset to include a wider variety of lesion types, stages, and patient demographics, ensuring a more comprehensive and inclusive model training process. Lastly, this study predominantly focuses on the model’s ability to generate explanations for medical practitioners, with less emphasis on patient comprehension. Developing patient-friendly explanation modules that translate complex diagnostic information into understandable insights could enhance patient engagement and satisfaction, paving the way for a more patient-centric approach in AI-assisted diagnostics.

## Figures and Tables

**Figure 1 diagnostics-14-00753-f001:**
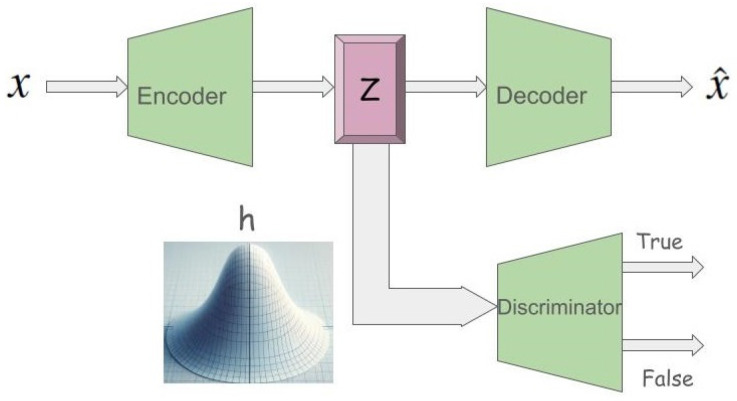
Adversarial autoencoder architecture.

**Figure 2 diagnostics-14-00753-f002:**
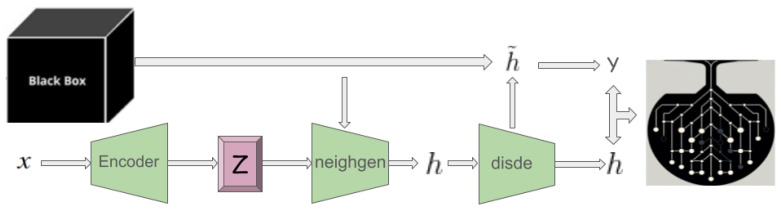
Latent local rules extractor (llore) module.

**Figure 3 diagnostics-14-00753-f003:**
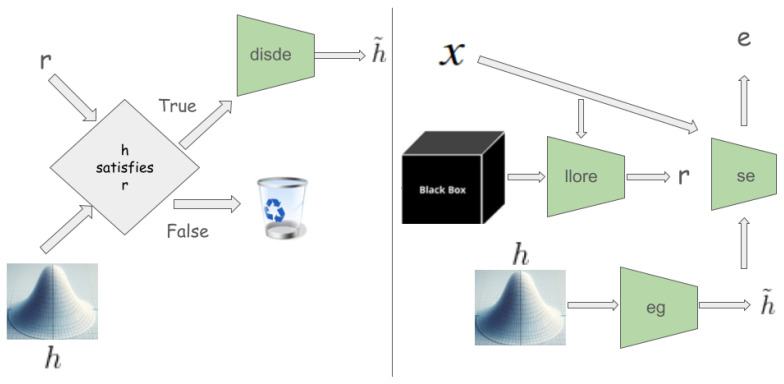
(**Left**): Exemplar generator (eg) module. (**Right**): abele architecture.

**Figure 4 diagnostics-14-00753-f004:**
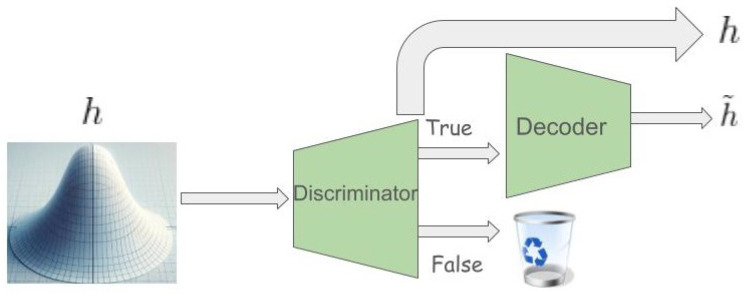
Discriminator and Decoder (disde) modules.

**Figure 5 diagnostics-14-00753-f005:**
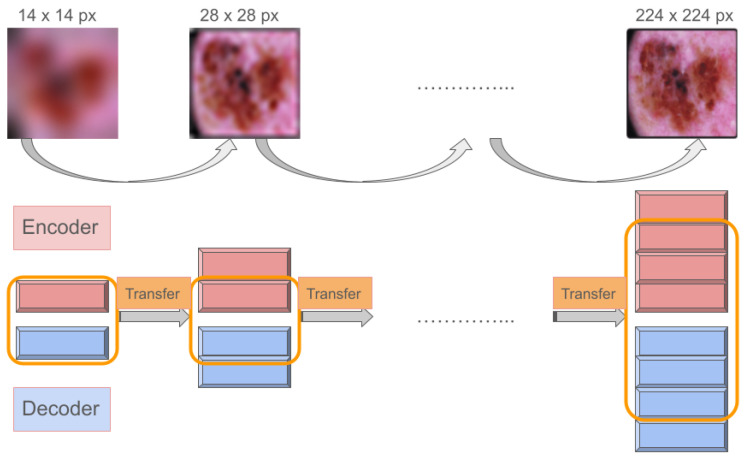
A Progressively Growing AAE. At each step, an autoencoder is trained to generate an image that is twice the size of the previous one, starting from an image of 14 × 14 pixels and gradually increasing to an image of 224 × 224 pixels. The learned features from one autoencoder are then transferred to the next. To handle the growing image size, both the encoder and decoder networks are expanded by adding one convolutional block at each step. The transfer learning is confined to the shared network architecture.

**Figure 6 diagnostics-14-00753-f006:**
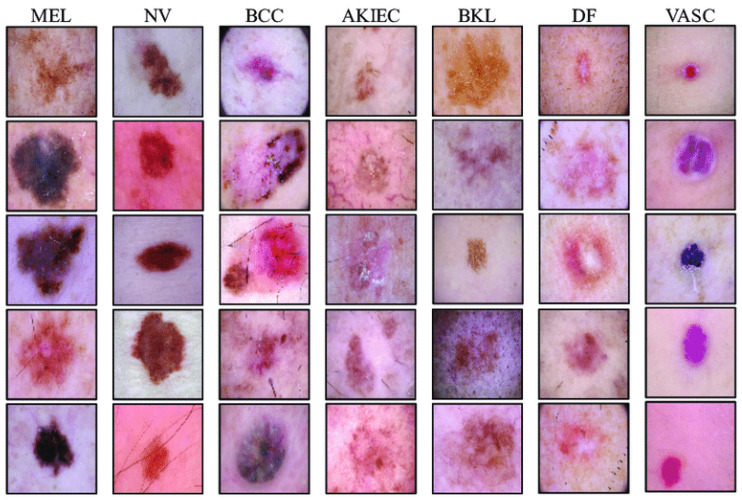
Dermoscopic images sampled from ISIC 2019 dataset.

**Figure 7 diagnostics-14-00753-f007:**
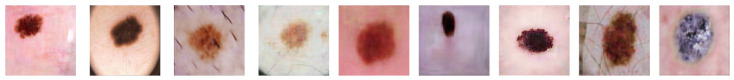
Synthetic skin lesion samples generated by abele and classified as melanocytic nevus by the ResNet black box, except for the upper-right image classified as actinic keratosis.

**Figure 8 diagnostics-14-00753-f008:**
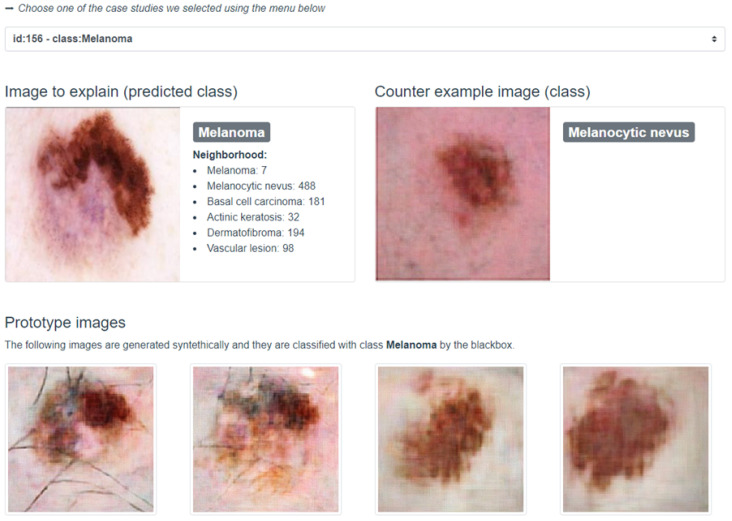
User visualization module to present the classification and the corresponding explanation. The upper part presents the input instance and a counter-exemplar. The lower part shows four exemplars that share the same class as the input.

**Figure 9 diagnostics-14-00753-f009:**
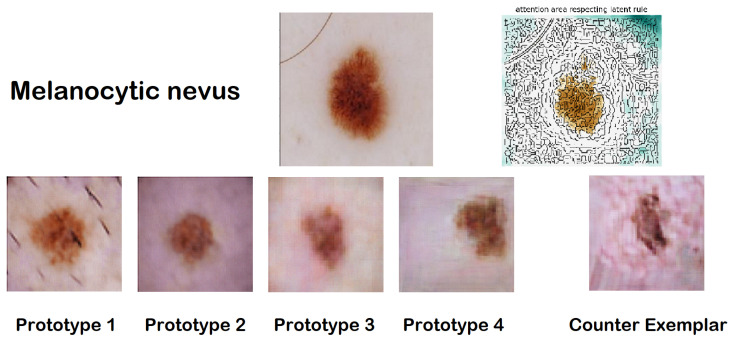
ABELE graphic explanation for a melanocytic nevus.

**Figure 10 diagnostics-14-00753-f010:**
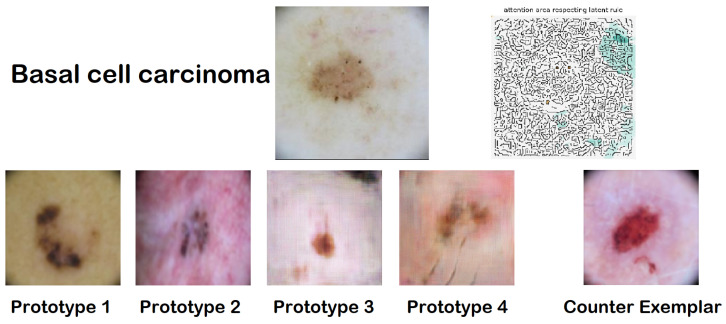
ABELE graphic explanation for a basal cell carcinoma.

**Figure 11 diagnostics-14-00753-f011:**
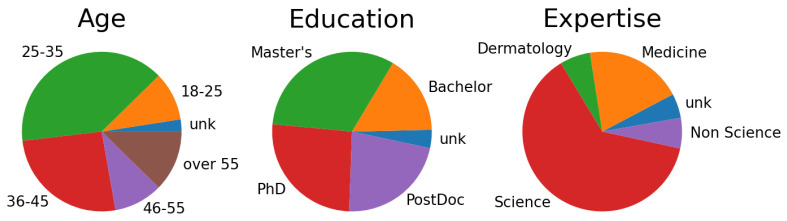
Demographic statistics of the survey participants.

**Figure 12 diagnostics-14-00753-f012:**
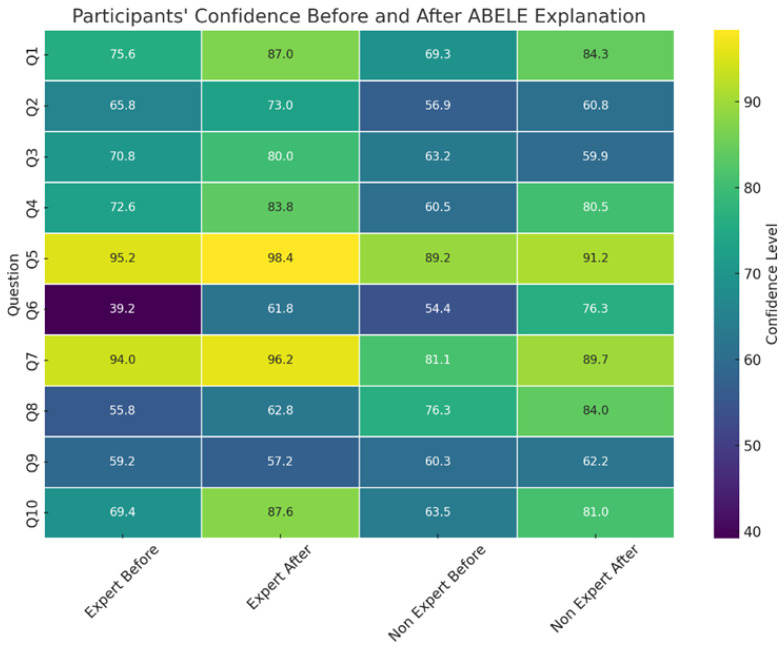
Participants’ confidence in the classification of the black box before and after receiving the explanation of abele.

**Figure 13 diagnostics-14-00753-f013:**
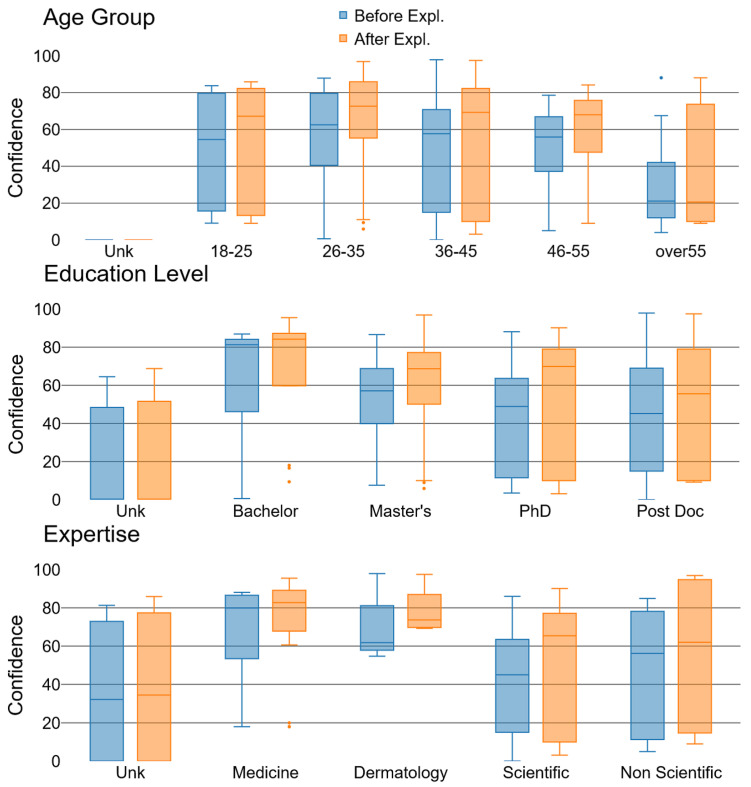
Participants’ confidence among different age groups (**top**), education level (**center**), domains (**bottom**), before and after explanations (from [17]).

**Figure 14 diagnostics-14-00753-f014:**
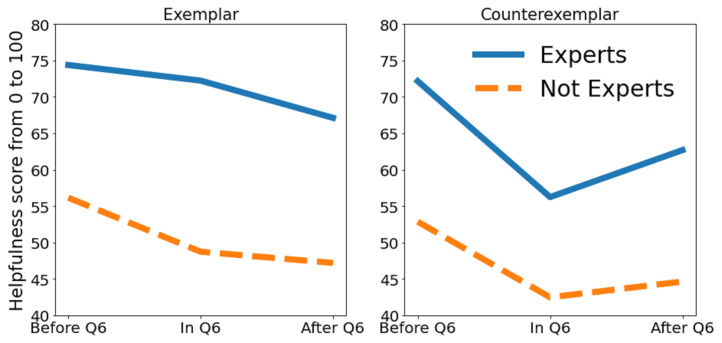
How much exemplars and counter-exemplars helped according to the participants’ responses, divided between groups of experts and non-experts (from [17]).

**Figure 15 diagnostics-14-00753-f015:**
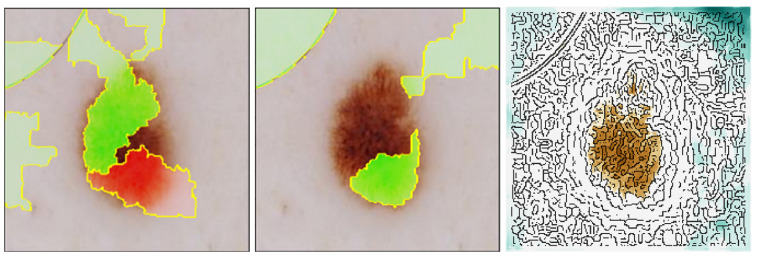
Saliency maps for lime (**left**), lore (**center**), and abele (**right**). lime and lore highlight the macro-regions of the image that contribute positively (green) or negatively (red) to the prediction while abele provides a more fine-grained level of information with a divergent color scale, from relevant areas (dark orange) to marginally significant areas (green/cyan) (from [17]).

**Figure 16 diagnostics-14-00753-f016:**
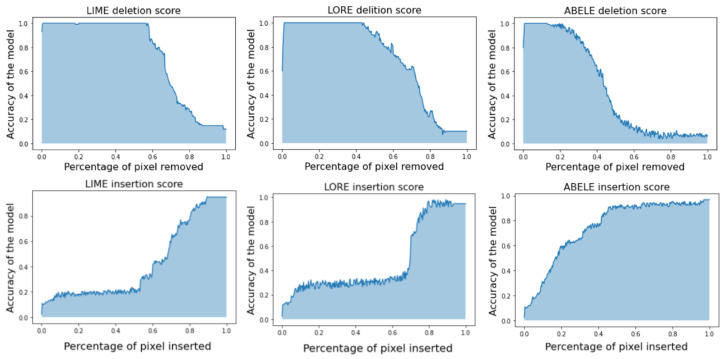
Deletion (**Top**) and Insertion (**Bottom**) metrics expressed as mean AUC of accuracy vs. percentage of removed or inserted pixels for 200 sample images. abele deletion curve drops earlier and faster relative to the percentage of removed pixels, signaling finer and more granular maps. abele insertion curve grows much earlier with respect to lime and lore (from [17]).

**Figure 17 diagnostics-14-00753-f017:**
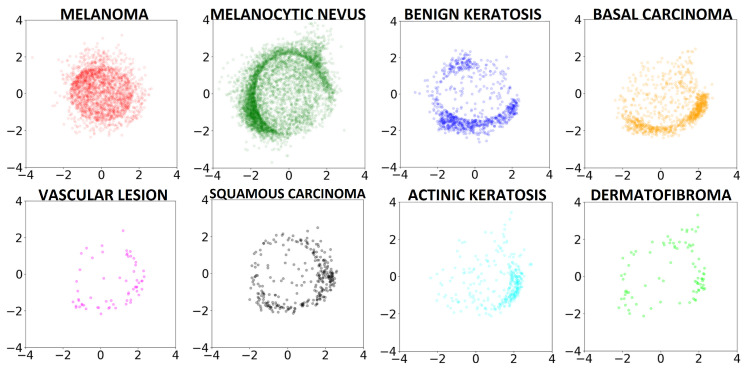
Training set represented in two dimensions through an MDS applied on the latent space learned by the PGAAE (from [17]).

**Figure 18 diagnostics-14-00753-f018:**
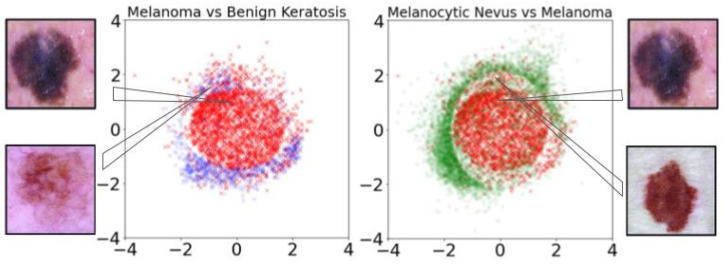
Visual separation between melanoma and benign keratosis (**Left**) and melanocytic nevus (**Right**).

**Figure 19 diagnostics-14-00753-f019:**
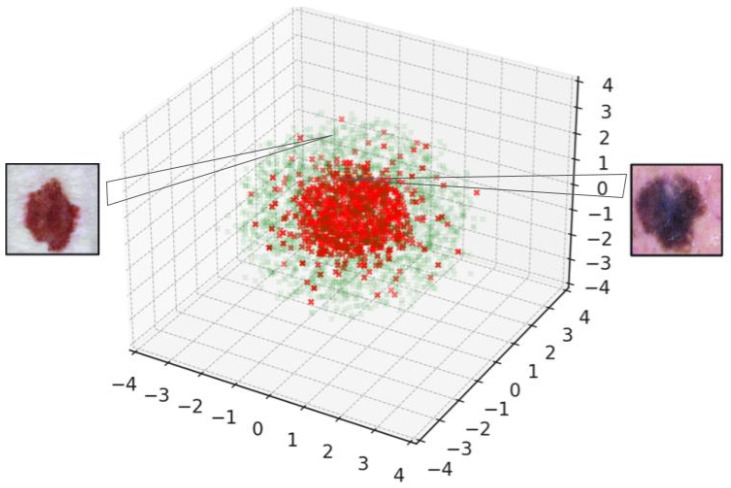
Visual 3d separation between melanoma (red) and melanocytic nevus (green).

**Table 1 diagnostics-14-00753-t001:** Detailed Black Box performance on each skin lesion category.

Metrics	Diagnosis Categories							
	SCC	MEL	NV	BCC	AK	BKL	DF	VASC
Recall	0.836	0.818	0.927	0.942	0.890	0.847	0.963	0.886
Specificity	0.969	0.926	0.882	0.960	0.964	0.955	0.988	0.996
Accuracy	0.966	0.906	0.906	0.958	0.961	0.944	0.987	0.995
F1	0.505	0.763	0.910	0.845	0.624	0.753	0.630	0.772
PPV	0.362	0.716	0.894	0.766	0.480	0.678	0.469	0.684
NPV	0.996	0.957	0.918	0.991	0.995	0.982	0.999	0.999
AUC	0.978	0.948	0.967	0.990	0.980	0.966	0.996	0.997
AUC80	0.967	0.907	0.940	0.984	0.970	0.946	0.996	0.997
AP	0.810	0.844	0.967	0.941	0.802	0.848	0.910	0.917

## Data Availability

ISIC 2019 dataset is publicly available at https://challenge.isic-archive.com/data (accessed on 1 September 2019).

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
