# Peer review of "Advancing Dermatological Diagnostics: Interpretable AI for Enhanced Skin Lesion Classification"

_diagnostics, 2024, doi:10.3390/diagnostics14070753_

Round 1

Reviewer 1 Report

Comments and Suggestions for Authors

Excellent paper.

One question that can be addressed in minor revision

Lines 380-382. Case study uses a set of images from a competition. Did the authors use the 25,331 labeled images in the original training set or include the additional 8328 unlabeled images in the original test set? If so how was the ground truth established? (I assume that they used the labeled images)

Authors excluded the "unknown" images. How many images were left for the training and validation datasets?

Please provide a confusion matrix for the classifier showing accuracy of classification for all 9 conditions

Not clear from the present text what the questions (Q1..Q10) were. Please include a table listing them. 

Excellent paper!!

Author Response

Thank you for the effort in reviewing our manuscript, Advancing Dermatological Diagnostics: Interpretable AI for Enhanced Skin Lesion Classification. In the attached file we address, one by one, the concerns of the reviewers, hoping that our efforts can clarify and improve our contribution to a level that you deem acceptable.

Yours sincerely,

The Authors.

Reviewer 2 Report

Comments and Suggestions for Authors

Dear Authors,

First of all, I congratulate you for your work. I have completed my evaluation of your work. I have indicated below the areas that I see necessary in the study. I hope that making the revisions specified in these items will contribute to your work. I wish you success.

Evaluation

-              There are punctuation errors in different parts of the work. It would be useful to review it in its entirety.

-              Skin lesion studies are very common recently. However, the literature of this study is limited. References can be expanded. 

-              The study has 8% similarity with only one publication. This rate should be reduced even if the authors have their own publications.

-              When applying the ResNet model, 80%-20% of the dataset is partitioned. Applying a k-fold cross validation technique could have increased the reliability of the results in the study. Instead, what procedures were performed for reliability?

-              Are there any limitations of the study? If so, it would be useful to mention them in the Conclusion section.

Author Response

(The authors gave the same response as above.)
